# A LATENT SPACE THEORY FOR EMERGENT ABILITIES IN LARGE LANGUAGE MODELS

## ABSTRACT

Languages are not created randomly but rather to communicate information. There is a strong association between languages and their underlying meanings, resulting in a sparse joint distribution that is heavily peaked according to their correlations. Moreover, these peak values happen to match with the marginal distribution of languages due to the sparsity. With the advent of LLMs trained on big data and large models, we can now precisely assess the marginal distribution of languages, providing a convenient means of exploring the sparse structures in the joint distribution for effective inferences. In this paper, we categorize languages as either unambiguous or $\varepsilon$-ambiguous and present quantitative results to demonstrate that the emergent abilities of LLMs, such as language understanding, in-context learning, chain-of-thought prompting, and effective instruction fine-tuning, can all be attributed to Bayesian inference on the sparse joint distribution of languages.

## 1 INTRODUCTION

Over the past few years, large language models (LLMs) have emerged as the predominant method for most natural language processing (NLP) tasks (Radford et al., 2018; Raffel et al., 2020; Brown et al., 2020; Chowdhery et al., 2022). With the increase in model size and training data, LLMs have demonstrated remarkable capabilities in solving various NLP tasks, including semantic understanding, few-shot in-context learning, chain-of-thought prompting, and effective instruction fine-tuning for alignment. These abilities are often referred to as emergent capabilities as they have been observed to emerge as the model size and training data increase (Wei et al., 2022b). In contrast to a simple scaling law, they are not the same abilities just extended to a new data distribution but some new abilities unseen in smaller model/data scales. Machine learning researchers are keen to know how LLMs have developed these skills to perform well on unseen tasks, especially since LLMs are primarily trained in an unsupervised manner to predict the next tokens in text. Some empirical studies have suggested that the emergent abilities of LLMs may be linked to the label space and input data distributional properties (Min et al., 2022; Chan et al., 2022), multitask prompted learning (Sanh et al., 2022), and pre-training term frequencies (Razeghi et al., 2022). Moreover, Xie et al. (2022); Wang et al. (2023) have proposed theories that explain in-context learning of LLMs as Bayesian inferences that use prompts to recover latent concepts. More recently, Wies et al. (2023) established a PAC based framework for in-context learnability, and Hahn and Goyal (2023) claimed that the emergent abilities arise through recombination of compositional structures in natural languages.

Motivated by Xie et al. (2022), our study proposes a novel latent space theory to explain the emergent abilities of LLMs. While Xie et al. (2022) considered a specific type of data distribution generated by Hidden Markov Models (HMMs), we examine general data distributions by exploring the sparsity property that is universally present in the joint distributions of languages. LLMs, which serve as a universal density approximator to the marginal distribution, offer a convenient means of exploring these sparse structures for effective inferences. We categorize languages as either unambiguous or $\varepsilon$-ambiguous and present quantitative results demonstrating that the emergent abilities of LLMs can be attributed to Bayesian inference on the sparse joint distribution of languages. Furthermore, we provide simulation results on synthetic languages that validate our theoretical findings.

## 2 A LATENT SPACE MODEL FOR LANGUAGE GENERATION

Languages are not created randomly, but with a specific purpose in mind, which is to convey information. Languages are composed of distinct, relatively independent units, such as sentences in natural languages or statements in programming languages. These separate pieces of language are referred to as *"messages"* in this paper. Each message, represented as $\mathbf{x}$, is in turn composed of a sequence of symbols from an alphabet with varying lengths. A message is created with the aim of expressing a single and definite intention, denoted as $\theta$. The set of all possible intentions constitutes another space, denoted as $\Theta$. We assume that the intention space $\Theta$ is a countable set of many distinct intentions. Each $\theta$ may represent a simple intention, which is an element from a finite set, or a composite intention that is made up of several simpler concepts or components through concatenation or recursion. Here we only require that the intention space $\Theta$ is discrete and complete, and each element in $\Theta$ is unique.

In this study, we investigate a popular stochastic process for language generation (Pieraccini and Levin, 1992; Miller et al., 1994):

**Step one**: To generate a message, we first select an intention $\theta$ we wish to convey from a prior distribution $q(\theta)$: $\theta \sim q(\theta)$. Since the underlying intention $\theta$ is not explicitly specified, it is considered a latent variable, and therefore the intention space $\Theta$ is a latent space.

**Step two**: After sampling an intention, say $\theta_0$, from the prior distribution in the previous step, we generate a message $\mathbf{x}$ to convey the intention using a conditional distribution $q(\mathbf{x}|\theta)$ by conditioning on $\theta_0$, i.e., $\mathbf{x} \sim q(\mathbf{x}|\theta_0)$. To ensure that the generated message is meaningful, we require that the conditional distribution $q(\mathbf{x}|\theta)$ is appropriately formulated such that the generated messages satisfy one of the following conditions:

1. **Unambiguous condition**:
$$\Pr(\theta_0|\mathbf{x}) = 1. \tag{1}$$
   This condition implies that we can precisely infer the underlying intention from the message with certainty. If this condition holds for all messages generated in a language, it is called an *unambiguous language*.

2. **Dominant condition with $\varepsilon$-ambiguity**:
$$\Pr(\theta_0|\mathbf{x}) \geq 1 - \varepsilon(\mathbf{x}) \quad (\text{with } 0 \leq \varepsilon(\mathbf{x}) < 1). \tag{2}$$
   In this scenario, we cannot definitively deduce the underlying intention from the observed message, but we can do so with a reasonable level of confidence. If every message generated in a language satisfy this condition with an $\varepsilon \in [0, 1)$, it is called an $\varepsilon$-ambiguous language. Each generated message has an $\varepsilon$ value that measures its degree of ambiguity. The smaller the $\varepsilon$ value of a message, the more precise and explicit it is. For any given intention $\theta_0$, it is possible to revise the message $\mathbf{x}$ in order to reduce its $\varepsilon$ value. This can be achieved by adding more details or choosing a better expression that makes the message less ambiguous.

From the perspective of communication, it makes sense that any practically-useful languages have been designed or evolved to satisfy either the condition of being unambiguous or the condition of being dominant. Otherwise, a message $\mathbf{x}$ becomes gibberish so that it does not serve the purpose of communication. It is important to note that neither of these conditions limits the possibility of expressing a single intention through multiple messages. The key constraint is whether or not we can confidently infer the intended meaning from each given message. For example, computer languages are designed to satisfy the unambiguous condition. For any computation task (viewed as an intention $\theta_0$), we can always write many different programs to achieve the same purpose. However, once we are given a piece of computer programs, it will perform one particular computation without any ambiguity. Even if the program has errors or bugs, it still carries out a specific action in a definitive manner. On the other hand, natural languages are well known to be notoriously ambiguous. Almost every message written in a natural language can be misinterpreted one way or another. However, we argue that every meaningful message in natural languages must satisfy the dominant condition so that the probability of misunderstanding is bounded by a sufficiently small number. If the probability of miscommunication is kept low, the message can still be valuable in conveying some information. In such cases, generating additional messages that are aligned with the same intention can enhance the likelihood of effectively communicating the intended meaning to a level that is considered adequate.

**Proposition 1** *Assume that two $\varepsilon$-ambiguity messages, i.e. $\mathbf{x}_1$ and $\mathbf{x}_2$, are independently generated under a common intention $\theta_*$, and their ambiguity levels are $\varepsilon_1$ and $\varepsilon_2$ respectively. If these two independent messages are concatenated as a single composite message $(\mathbf{x}_1, \mathbf{x}_2)$, the level of ambiguity in determining the shared intention $\theta_*$ decreases to $\varepsilon_1\varepsilon_2$.* [1]

The joint distribution of languages, whether they are unambiguous or $\varepsilon$-ambiguous, shows an intriguing property of sparsity (see Appendix B). For any unambiguous language, we have

$$\Pr(\theta_0|\mathbf{x}) = 1 \implies q(\theta, \mathbf{x}) = \begin{cases} q(\mathbf{x}) & \theta = \theta_0 \\ 0 & \theta \neq \theta_0 \end{cases} \tag{3}$$

For an $\varepsilon$-ambiguous language, we have

$$\Pr(\theta_0|\mathbf{x}) \geq 1 - \varepsilon(\mathbf{x}) \implies \frac{q(\theta_0, \mathbf{x})}{q(\Theta\backslash\theta_0, \mathbf{x})} \geq \frac{1}{\varepsilon(\mathbf{x})} \tag{4}$$

where $\Theta\backslash\theta_0$ denotes a complement set containing all elements in $\Theta$ except $\theta_0$, and $q(\Theta\backslash\theta_0, \mathbf{x})$ is defined as the total residue after the contribution of the intention $\theta_0$ is taken out: $q(\Theta\backslash\theta_0, \mathbf{x}) \triangleq \sum_{\theta\in\Theta,\theta\neq\theta_0} q(\theta, \mathbf{x}) = q(\mathbf{x}) - q(\theta_0, \mathbf{x})$.

The above latent space model can be extended to generate a composite segment of languages that is composed of multiple messages, i.e., $\mathbf{X} = \{\mathbf{x}_1, \mathbf{x}_2, \cdots, \mathbf{x}_m\}$. The marginal distribution of $\mathbf{X}$ can be computed based on the latent space model as:

$$q(\mathbf{X}) = q(\mathbf{x}_1, \mathbf{x}_2, \cdots, \mathbf{x}_m) = \sum_{\theta_1, \cdots, \theta_m} q(\theta_1, \theta_2, \cdots, \theta_m) \prod_{i=1}^{m} q(\mathbf{x}_i|\theta_i)$$

where $\theta_i$ ($\forall i = 1, 2, \cdots, m$) denotes the latent variable under which each message $\mathbf{x}_i$ is generated. If these intentions are coherent, their joint probability $q(\theta_1, \theta_2, \cdots, \theta_m)$ is high, otherwise it is low. Obviously, for unambiguous or $\varepsilon$-ambiguous languages, the term corresponding to the actual intentions of all messages will dominate the above summation over other possible intentions.

The above latent space model for language generation has been known for a long time (Pieraccini and Levin, 1992; Miller et al., 1994; Blei et al., 2003; Wei et al., 2022a). However, when we apply this model to any practical language-related task, we face two insurmountable challenges:
*i) The latent space $\Theta$ is unknown.* We can only be certain that $\Theta$ is a complete set encompassing all possible distinct intentions that are relevant to the underlying task. We lack understanding of what the individual components of $\theta$ may be and how to effectively construct them in order to achieve a complete $\Theta$ for any practical application. For a considerable period of time, many AI researchers have maintained the belief that we cannot achieve artificial general intelligence without possessing the full knowledge and ability to explicitly specify and model the underlying latent space $\Theta$.
*ii) All true probability distributions, i.e. $q(\theta)$, $q(\mathbf{x}|\theta)$, $q(\mathbf{x})$, are not available.* In the past, a variety of statistical models have been proposed to approximate these unknown distributions. However, due to the constraints in computing resources, it tended to use simple or small models for this purpose, which could only provide rough estimates for these true distributions and were never able to assess them with precision.

This paper aims to elucidate why and how the existing practices of utilizing LLMs have inadvertently addressed the two aforementioned issues, leading to a significant enhancement in the inference performance without requiring explicit knowledge of $\Theta$. This sudden improvement is often viewed as the emergent abilities from these extensive language models as they appears simply as a result of increasing model scale and data size.

## 3 LLMs are a universal density approximator of marginal distribution $q(\mathbf{x})$

For any message $\mathbf{x}$ in a language consisting of $T$ symbols, $\mathbf{x} = \{x_1, x_2, \cdots, x_T\}$, according to the general product rule of probability, we have

$$q(\mathbf{x}) = q(x_1, x_2, \cdots, x_T) = q(x_1)q(x_2|x_1)\cdots q(x_T|x_1, x_2, \cdots, x_{T-1}).$$

---

[1]See a detailed proof in Appendix A.

If we can construct a statistical model to accurately compute these conditional probabilities conditioned on any history string (no more than $T-1$ symbols), denoted as $\mathbf{h}$ for notational convenience, we will be able to compute the true marginal distribution $q(\mathbf{x})$ for any message of no more than $T$ symbols. Let's denote this statistical model as $p_{\boldsymbol{\Lambda}}(x|\mathbf{h})$, where $\boldsymbol{\Lambda}$ stands for all parameters of this model, we may use this model to compute a distribution for any $\mathbf{x}$ (no more than $T$ symbols) as follows:

$$p_{\boldsymbol{\Lambda}}(\mathbf{x}) = p_{\boldsymbol{\Lambda}}(x_1, x_2, \cdots, x_T) = p_{\boldsymbol{\Lambda}}(x_1)p_{\boldsymbol{\Lambda}}(x_2|x_1)\cdots p_{\boldsymbol{\Lambda}}(x_T|x_1, x_2, \cdots, x_{T-1})$$

Generally speaking, $p_{\boldsymbol{\Lambda}}(\mathbf{x}) \neq q(\mathbf{x})$ because the statistical model can only be viewed as a rough approximation of the true marginal distribution $q(\mathbf{x})$ at best. In the following, we will discuss a special case where a sufficiently large statistical model is constructed so that $p_{\boldsymbol{\Lambda}}(\mathbf{x})$ can approach $q(\mathbf{x})$ asymptotically.

**Definition 1** *A parametric statistical model $p_{\boldsymbol{\Lambda}}(\mathbf{x})$ is called a universal density approximator if there exists a set of parameters $\boldsymbol{\Lambda}$ for any given probability distribution $q(\mathbf{x})$ so that $p_{\boldsymbol{\Lambda}}(\mathbf{x})$ can approximate $q(\mathbf{x})$ up to any precision:*

$$\forall q(\mathbf{x}), \forall \epsilon > 0, \exists \boldsymbol{\Lambda}: \quad \left| q(\mathbf{x}) - p_{\boldsymbol{\Lambda}}(\mathbf{x}) \right| \leq \epsilon \quad (\forall \mathbf{x}).$$

**Lemma 1** *A composition of some universal function approximators and the softmax function leads to a universal density approximator to the above conditional distribution $q(x|\mathbf{h})$.*

For any history $\mathbf{h}$, $q(x|\mathbf{h})$ is a multinomial distribution (of all unique symbols in alphabet), we can take logarithm to obtain the required input for the softmax function to derive this multinomial distribution. A set of universal function approximators can be constructed to map any $\mathbf{h}$ to its required input so that the given conditional distribution is modelled precisely (Asadi and Jiang, 2020; Jiang, 2021).

**Lemma 2** *A sufficiently large transformer model is a universal function approximator.*

Yun et al. (2020) have proved that transformer models (Vaswani et al., 2017) can universally approximate arbitrary continuous sequence-to-sequence functions. An immediate corollary is that transformers can precisely map each input sequence $\mathbf{h}$ to any output values.

**Theorem 1** *If a statistical model $p_{\boldsymbol{\Lambda}}(\mathbf{x})$ is a universal density approximator, the model parameters $\boldsymbol{\Lambda}_n$ is the maximum likelihood estimation from a training set consisting of $n$ i.i.d. samples randomly drawn from any distribution $q(\mathbf{x})$, then we have*

$$\lim_{n \to \infty} p_{\boldsymbol{\Lambda}_n}(\mathbf{x}) = q(\mathbf{x})$$

*for all $\mathbf{x}$.*

Theorem 1 can be viewed as a corollary from the consistency property of maximum likelihood estimation (Lehmann and Casella, 1998; Hogg et al., 2019). See Appendix C for a proof.

As in today's large language models (LLMs), if we construct a universal density approximator $p_{\boldsymbol{\Lambda}}(x|\mathbf{h})$ for any history $\mathbf{h}$ (no more than $T-1$ symbols) using the so-called GPT structures (Radford et al., 2018; Brown et al., 2020), and furthermore if we use maximum likelihood estimation to derive its parameters, denoted as $\boldsymbol{\Lambda}_*$, from an infinite number of text messages that are presumed to be generated from the above latent space model (as in Brown et al. (2020)), according to Lemmas 1 and 2, and Theorem 1, we have

$$p_{\boldsymbol{\Lambda}_*}(\mathbf{x}) = q(\mathbf{x}) \tag{5}$$

for all $\mathbf{x}$ (no more than $T$ symbols) generated from the same latent space model. In other words, today's LLM techniques allow us to accurately assess the margin distribution of languages, which was not possible in the past. In Merrill et al. (2022), a similar assertion is put forth, suggesting that an ideal language model, having thoroughly learned its intended distribution, can effectively establish the entailment relationship between sentences.

## 4    LANGUAGE UNDERSTANDING WITH LLMS

Large Language models (LLMs) have surprised many NLP researchers with their ability to compre­hend any given text prompt and generate highly relevant fluent responses. What is remarkable is that LLMs are trained through a predict-next-token approach, which is not explicitly linked with the semantic meanings of languages. However, it is easy to tell that this predict-next-token approach is essentially the maximum likelihood estimation of the marginal distribution $q(\mathbf{x})$. Assume an LLM $\mathbf{\Lambda}_*$ is well trained from an infinite number of text messages so that this model converges to the true marginal distribution of languages as shown in eq.(5). Despite this seemingly limited training approach, LLMs demonstrate a remarkable capacity to understand the content of any text prompt and generate coherent and appropriate responses. Here, we will explain how this happens in LLMs.

Assume the prompt $\mathbf{x}$ is a message generated from the above latent-space-based language generation process, where $\mathbf{x}$ is generated based on a hidden intention $\theta_{\mathbf{x}}$. We do not know exactly what $\theta_{\mathbf{x}}$ is but we are sure it belongs to the latent space $\mathbf{\Theta}$. For simplicity, we first assume the language generation process is unambiguous. When we sample the LLM $\mathbf{\Lambda}_*$ to generate a reply conditioned on the prompt $\mathbf{x}$, we can easily compute the distribution of all possible replies $\mathbf{y}$ as follows:

$$p_{\mathbf{\Lambda}_*}(\mathbf{y}|\mathbf{x}) = \frac{p_{\mathbf{\Lambda}_*}(\mathbf{x}, \mathbf{y})}{p_{\mathbf{\Lambda}_*}(\mathbf{x})} = \frac{q(\mathbf{x}, \mathbf{y})}{q(\mathbf{x})}$$

Let's use $\theta$ to indicate the latent variable in the above language generation process, we further derive the numerator as follows:

$$q(\mathbf{x}, \mathbf{y}) = \sum_{\theta \in \mathbf{\Theta}} q(\mathbf{x}, \theta, \mathbf{y}) = \sum_{\theta \in \mathbf{\Theta}} q(\mathbf{x}, \theta) \, q(\mathbf{y}|\mathbf{x}, \theta) = q(\mathbf{x}) \, q(\mathbf{y}|\mathbf{x}, \theta_{\mathbf{x}})$$

The last step in the above relies on the sparsity property of unambiguous languages in (3). Substituting it to the above equation, we further have

$$p_{\mathbf{\Lambda}_*}(\mathbf{y}|\mathbf{x}) = q(\mathbf{y}|\mathbf{x}, \theta_{\mathbf{x}})$$

In other words, when we sample the LLMs to generate a reply based on the prompt $\mathbf{x}$, it essentially uses the exactly same language generation process conditioned on both the prompt and its true unknown intention $\theta_{\mathbf{x}}$. Even though we do not know what $\theta_{\mathbf{x}}$ is, the magic of the LLMs lies at that this unknown intention can be implicitly explored by the LLMs in such a way that the reply $\mathbf{y}$ is generated under this intention to continue the prompt $\mathbf{x}$. This is the fundamental reason why the LLMs can perfectly understand the meaning of any text prompt and accordingly generate a highly relevant and fluent reply because the conditional sampling on LLMs is essentially identical to the same language generation process that yields a text message under the intention $\theta_{\mathbf{x}}$ in the first place.

Next, let's extend the above result to $\varepsilon$-ambiguous languages.

**Proposition 2** *For $\varepsilon$-ambiguous languages, the conditional distribution of LLMs and the true condi­tional distribution in the language generation process are bounded as:*

$$\left| p_{\mathbf{\Lambda}_*}(\mathbf{y}|\mathbf{x}) - q(\mathbf{y}|\mathbf{x}, \theta_{\mathbf{x}}) \right| \leq \varepsilon(\mathbf{x}) \tag{6}$$

*where $\varepsilon(\mathbf{x})$ denotes the ambiguity of the prompt.*

See Appendix D for a detailed proof. Proposition 2 suggests that the ambiguity in language generation deviates the distribution of all possible replies from the ideal distribution in the language generation. However, their difference is bounded by the ambiguity of the prompt, i.e. $\varepsilon(\mathbf{x})$. This explains why choosing a well-crafted prompt can minimize the margin of error and result in more proficient responses (Liu et al., 2021; Rubin et al., 2022).

## 5    IN-CONTEXT LEARNING IN LLMS

One impressive capability of large language models (LLMs) is their ability to perform few-shot in-context learning. Even when presented with tasks that were not included in their initial training, LLMs can quickly learn these tasks by observing just a few examples without any need to modify

their model parameters. After observing a few such examples, the LLM can immediately apply the newly acquired skill to any new cases following the examples.

Let's first explain the typical setting for few-shot in-context learning in LLMs. Assume we want to teach a pre-trained LLM $\mathbf{\Lambda}_*$ to conduct a new task by providing an instruction text $\mathbf{x}$, along with a few independent examples of input-output pairs associated with this new task, i.e. $(\mathbf{i}_1, \mathbf{o}_1), (\mathbf{i}_2, \mathbf{o}_2), \cdots, (\mathbf{i}_m, \mathbf{o}_m)$. We further assume that the underlying unknown intention associated with this task is denoted as $\theta_*$. In other words, both the instruction $\mathbf{x}$ and all provided examples are generated independently under the same intention $\theta_*$ of this task (Xie et al., 2022). Moreover, since these examples correctly reflect how this task is performed, we have $q(\mathbf{o}_k \,|\, \mathbf{i}_k, \theta_*) \approx 1$ ($\forall k = 1, 2, \cdots, m$).

We have the following in-context learning results for unambiguous languages and $\varepsilon$-ambiguous languages, respectively.

**Proposition 3** *Under the above setting, for unambiguous languages, the conditional probability distribution of the LLM $\mathbf{\Lambda}_*$ given the instruction prompt and all examples and a new case is equal to the true conditional distribution used in language process under the common intention $\theta_*$:*

$$p_{\mathbf{\Lambda}_*}(\mathbf{y}|\mathbf{x}, \mathbf{i}_1, \mathbf{o}_1, \cdots, \mathbf{i}_m, \mathbf{o}_m, \mathbf{i}_{m+1}) = q(\mathbf{y}|\mathbf{i}_{m+1}, \theta_*) \tag{7}$$

See a detailed proof in Appendix E. Proposition 3 suggests that, if we sample the conditional distribution of the LLM, it is equivalent to sampling the ideal conditional distribution in language generation under the shared true intention $\theta_*$ and the new input $\mathbf{i}_{i+1}$. As this conditional distribution peaks at the correct output $\mathbf{o}_{m+1}$, i.e. $q(\mathbf{o}_{m+1} \,|\, \mathbf{i}_{m+1}, \theta_*) \approx 1$, we will get $\mathbf{o}_{m+1}$ at a high probability. Moreover, as we have seen at above, for unambiguous languages, it does not help much by providing more examples for in-context learning. Next, let's see what may happen if we perform in-context learning on $\varepsilon$-ambiguous languages.

**Proposition 4** *Under the same setting, for $\varepsilon$-ambiguous languages, we have*

$$\left| p_{\mathbf{\Lambda}_*}(\mathbf{y}|\mathbf{x}, \mathbf{i}_1, \mathbf{o}_1, \cdots, \mathbf{i}_m, \mathbf{o}_m, \mathbf{i}_{m+1}) - q(\mathbf{y}|\mathbf{i}_{m+1}, \theta_*) \right| \le \varepsilon(\mathbf{x})\varepsilon(\mathbf{i}_{m+1}) \prod_{k=1}^{m} \varepsilon(\mathbf{i}_k, \mathbf{o}_k) \le \varepsilon_0^{m+2} \tag{8}$$

*where $\varepsilon(\mathbf{x})$ is the ambiguity of the prompt, $\varepsilon(\mathbf{i}_k, \mathbf{o}_k)$ the ambiguity of $k$-th example, and $\varepsilon(\mathbf{i}_{m+1})$ for $\mathbf{i}_{m+1}$. Here $\varepsilon_0$ stands for the maximum ambiguity among the prompt and all examples.*

See a detailed proof in Appendix F. Proposition 4 shows that LLMs deviate from the ideal generation process due to the language ambiguity but the error bound exponentially decreases with respect to the number of provided examples. As a result, for ambiguous languages, we tend to have better performance if more relevant examples are provided for few-shot in-context learning. This is consistent with what has been observed in many in-context learning experiments on LLMs (Brown et al., 2020).

## 6 CHAIN-OF-THOUGHT PROMPTING AND FINE-TUNING IN LLMS

If we are given a composite argument $\mathbf{X}$, consisting of a sequence of multiple messages as $\mathbf{X} = \{\mathbf{x}_0, \mathbf{x}_1, \cdots, \mathbf{x}_m\}$, where these messages are generated under their own intentions, i.e., $\{\theta_0, \theta_1, \cdots, \theta_m\}$. If these messages in the argument are coherent as a sequence of reasoning steps, it represents a higher level chain of thought, characterised by the sequence of these underlying intentions: $\theta_0 \to \theta_1 \to \cdots \to \theta_m$. Empirical results have shown that it will significantly improve the chance for LLMs to correctly arrive at the final conclusion if these intermediate reasoning steps are explicitly specified in prompts, which is often called *chain-of-thought prompting*. Here let's use the latent space model to explain how the chain-of-thought prompting may help LLMs to correctly infer the final conclusion in multi-step reasoning tasks.

Similarly, we start with a simple case for unambiguous languages. Taking the above composite argument $\mathbf{X}$ as example, assume $\mathbf{x}_0$ is the question, and $\mathbf{x}_1, \cdots, \mathbf{x}_{m-1}$ denotes all intermediate reasoning steps, and $\mathbf{x}_m$ the final conclusion. If we only prompt the question without using chain-of-thought, the probability for an LLM to arrive at the correct conclusion is computed as:

$$p_{\mathbf{\Lambda}_*}(\mathbf{x}_m|\mathbf{x}_0) = \frac{q(\mathbf{x}_0, \mathbf{x}_m)}{q(\mathbf{x}_0)} = \frac{q(\theta_0, \theta_m)q(\mathbf{x}_0|\theta_0)q(\mathbf{x}_m|\theta_m)}{q(\theta_0)q(\mathbf{x}_0|\theta_0)} = q(\theta_m|\theta_0)q(\mathbf{x}_m|\theta_m) \tag{9}$$

If the argument from $\mathbf{x}_0$ to $\mathbf{x}_m$ requires many reasoning steps, the direct transition from the intention $\theta_0$ to $\theta_m$ may be rare and under-represented in the training corpus so that $q(\theta_m|\theta_0)$ is extremely small. If we directly prompt the LLMs as above, the chance to arrive at the correct conclusion is slim. On the other hand, if we use the chain-of-thought prompting, the conditional probability is computed as:

$$p_{\mathbf{\Lambda}_*}(\mathbf{x}_m|\mathbf{x}_0, \mathbf{x}_1, \cdots, \mathbf{x}_{m-1}) = \frac{q(\mathbf{x}_0, \mathbf{x}_1, \cdots, \mathbf{x}_{m-1}, \mathbf{x}_m)}{q(\mathbf{x}_0, \mathbf{x}_1, \cdots, \mathbf{x}_{m-1})} = q(\theta_m|\theta_0, \cdots, \theta_{m-1})q(\mathbf{x}_m|\theta_m)$$

As $\theta_m$ is an immediate reasoning step after $\theta_0 \to \theta_1 \cdots \to \theta_{m-1}$, the transition to $\theta_m$ occurs more often in the training corpus, as there are many other training examples that may follow the same reasoning chain. As a result, the transition probabilities $q(\theta_m|\theta_0, \theta_1, \cdots, \theta_{m-1})$ are higher since more information is conditioned. This makes it more likely for the LLM to arrive at $\mathbf{x}_m$ by going through $\theta_0 \to \theta_1 \cdots \to \theta_{m-1}$ than by transitioning directly from $\theta_0$ to $\theta_m$.

## 7 ALIGNING LLMs TO FOLLOW INSTRUCTION

Assume that an LLM $\mathbf{\Lambda}_*$ is prompted with an instruction $\mathbf{x}$, which is a message generated under an unknown intention $\theta_{\mathbf{x}}$. Let's consider the conditional distribution of all possible responses $\mathbf{y}$ that may be generated by the LLM under this instruction. For unambiguous languages, we have

$$p_{\mathbf{\Lambda}_*}(\mathbf{y}|\mathbf{x}) = \frac{q(\mathbf{x}, \mathbf{y})}{q(\mathbf{x})} = \frac{\sum_{\theta'} \sum_{\theta} q(\mathbf{x}, \theta', \mathbf{y}, \theta)}{\sum_{\theta'} q(\mathbf{x}, \theta')} = \sum_{\theta \in \mathbf{\Theta}} q(\theta|\theta_{\mathbf{x}})q(\mathbf{y}|\theta) \tag{10}$$

where $\theta'$ and $\theta$ denote the latent variables to generate the instruction and its response.

Similarly, for $\varepsilon$-ambiguous languages, we have

$$\left| p_{\mathbf{\Lambda}_*}(\mathbf{y}|\mathbf{x}) - \sum_{\theta \in \mathbf{\Theta}} q(\theta|\theta_{\mathbf{x}})q(\mathbf{y}|\theta) \right| \leq \varepsilon(\mathbf{x}) \tag{11}$$

where $\varepsilon(\mathbf{x})$ denotes the ambiguity of the instruction.

The results presented above demonstrate that when an LLM receives an instruction $\mathbf{x}$, all possible responses follow a mixture distribution. Each specific response is generated by first selecting an intention $\theta$ based on the transition probability from the prompt intention, i.e. $q(\theta|\theta_{\mathbf{x}})$. Then, a response $\mathbf{y}$ is generated by sampling the conditional distribution, i.e. $q(\mathbf{y}|\theta)$, to reflect the selected intention. Therefore, the underlying intention of any generated response largely depends on the transition probability from the prompt intention, i.e. $q(\theta|\theta_{\mathbf{x}})$. If the LLM's training set is sufficiently large, these transition probabilities tend to take higher values for those intentions strongly related to the prompt intention $\theta_{\mathbf{x}}$, whether through causation or mere association. However, some of these intentions may be harmful or offensive, so LLMs need to be fine-tuned to avoid generating hurtful responses (Ouyang et al., 2022).

To fine-tune an LLM, a number of responses to common prompts are manually labeled as good or bad by human annotators. These labeled responses are then used to train the LLM via reinforcement learning so that the probabilities of generating bad responses are significantly reduced. By adjusting a single transition probability $q(\theta|\theta_{\mathbf{x}})$, it is possible to improve not only one particular instruction $\mathbf{x}$ but also many other instructions generated under the same intention. This is why instruction fine-tuning for LLMs is often very effective (Ouyang et al., 2022; Chung et al., 2022).

## 8 SIMULATION EXPERIMENTS

In order to verify the theoretical results on $\varepsilon$-ambiguous languages, we need to create synthetic languages with explicitly known distributions, and a simple mechanism to control the level of ambiguity. In this paper, we propose using a doubly-embedded Markov chain model to generate both unambiguous and $\varepsilon$-ambiguous languages. To generate the synthetic languages, we begin by sampling an intention from a circular Markov chain of six states that moves in one direction: $\theta_0 \to \theta_1 \to \theta_2 \to \theta_3 \to \theta_4 \to \theta_5 \to \theta_0$. Each state corresponds to a distinct intention, and has a 50% chance of staying and a 50% chance of jumping to the next state. After sampling an intention, we use another intention-specific Markov chain to generate a message for that intention.

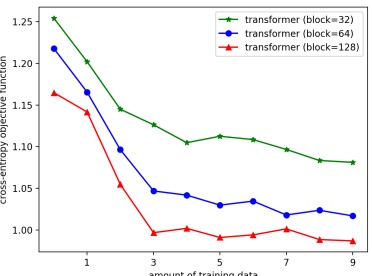 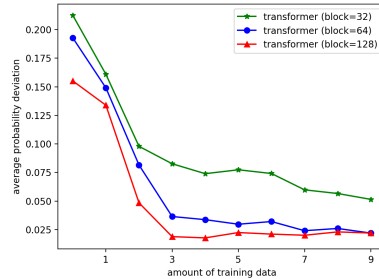

Figure 1: Illustration of the convergence of the transformer-based LLMs as more and more training data is added. Left: the cross-entropy objective functions. Right: The average difference between the probability distributions $p_{\Lambda_*}(\mathbf{x})$ and $q(\mathbf{x})$ on an unseen validation set.

These intention-dependent Markov chains consists of 18 states, corresponding to 18 alphabetic letters (from $a$ to $r$). To generate unambiguous languages, we use only three distinct letters to construct a message for each intention, such as using only $\{a, b, c\}$ for $\theta_0$, $\{d, e, f\}$ for $\theta_1$, and so on. In this case, each intention-dependent Markov chain has only $3 \times 3$ nonzero probabilities (corresponding to the three selected letters) in its $18 \times 18$ transition matrix. On the other hand, to generate $\varepsilon$-ambiguous languages, we inject small amounts of noise into these sparse transition matrices so that each letter has a small probability of jumping to other letters rather than those dedicated to the same intention. By varying the amount of noise added to the transition matrices, we can simulate different levels of $\varepsilon$-ambiguity. After generating a message for a given intention using an intention-dependent Markov chain, a *newline* character is appended to signify the message's end. After that, the next intention is selected using the circular Markov chain, and another message is generated for that intention in the same way. This process can be repeated as many times as desired to generate multiple messages. The true distributions of these generated languages can be precisely calculated from all transition matrices of the doubly-embedded Markov chain model. Therefore, these synthetic languages can be used in simulation experiments to verify the theoretical results on $\varepsilon$-ambiguous languages.

## 8.1 CONVERGENCE OF LLMS

For our experiments, we train a character-level GPT-like model (Radford et al., 2018) on the synthetic languages we generated. The GPT model has three layers and about 21 million parameters. We vary the amount of training data and the block size used by transformers in each layer. The convergence of transformer-based GPT models is depicted in Figure 1, as a function of the amount of training data. The models exhibit a rapid convergence on the simple synthetic language. The left panel of the figure shows the learning curves of three transformers of different block sizes, using the regular cross-entropy objective function (equivalent to the negative log-likelihood function). In the right panel, the average difference between the model distribution $p_{\Lambda_*}(\mathbf{x})$ and the true distribution $q(\mathbf{x})$ is shown when evaluated on an independent validation set. The results clearly indicate that the gap between these two probability distributions reduces quickly as the transformer models observe more training data. Interestingly, transformers model the true distribution slightly better with a larger block size, even though this does not increase the number of model parameters.

## 8.2 LANGUAGE UNDERSTANDING WITH LLMS

To validate Proposition 2, we first use the trained language model to generate a partial message $\mathbf{x}$ based on an intention $\theta_{\mathbf{x}}$. We then prompt the LLM with the partial message $\mathbf{x}$ and calculate the KL divergence between the model's conditional distribution $p_{\Lambda_*}(\mathbf{y}|\mathbf{x})$ and the true conditional distribution on the underlying intention $q(\mathbf{y}|\mathbf{x}, \theta_{\mathbf{x}})$. In the left of Figure 2, the average KL divergence of three languages with varying levels of ambiguity is displayed, demonstrating that the gap is small for unambiguous languages and slightly increases as the languages become more ambiguous.

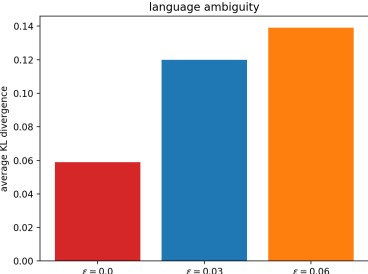 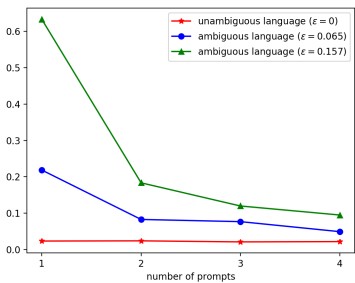

Figure 2: Illustration of language understanding (left) and in-context learning (right) of LLMs on synthetic languages with various levels of ambiguity.

## 8.3 IN-CONTEXT LEARNING IN LLMs

To verify the results presented in Propositions 3 and 4 regarding in-context learning, we follow a two-step process. Firstly, we generate multiple messages based on the same intention using the trained model. Then, we concatenate these messages to create a single prompt for the LLM. We then compare the KL divergence between the model's conditional distribution and the true distribution under the correct intention as a function of the number of independent messages in the prompt. The results, as shown in the right panel of Figure 2, demonstrate that the average KL divergence remains small and constant for unambiguous languages, but increases for $\varepsilon$-ambiguous languages. However, the gap for $\varepsilon$-ambiguous languages can be reduced by adding more messages to the prompt.

## 9 FINAL REMARKS

In this paper, we have presented a novel latent space theory that explains the remarkable capabilities of LLMs. Our theory categorizes languages as either unambiguous or $\varepsilon$-ambiguous, and establishes quantitative results that attribute the LLMs' emergent abilities to their unique approach for exploring sparse structures in joint distributions through being a universal approximator of the marginal distribution. Simulation experiments on synthetic data have confirmed our findings related to language ambiguity.

This paper assumes all languages are generated from a latent intention space. The key idea in this paper is that, without explicitly knowing how this intention space is constructed, LLMs can implicitly accessing the conditional distribution of text contingent on any intent in this space, provided that the model and data are sufficiently large. This elucidates the reason behind LLMs' ability to produce text resembling human replies, as these conditional distribution is essential in language generation, a process akin to human communication.

Our theoretical findings primarily revolve around an asymptotic analysis applied when a sufficiently expressive model is effectively trained on a substantial amount of data. In this scenario, eq.(5) holds rigorously, serving as the fundamental basis for all the results detailed in our paper. In practical settings, as we train increasingly larger models with progressively larger datasets, it is described by the limit in Theorem 1. This consistent convergence suggests that when we utilize more data and more capable models for training, the disparity between the two distributions in eq.(5) diminishes. As a result, we can draw the following conclusions: i) For small models, the gap between these distributions can be arbitrarily large, rendering the content discussed in our paper inapplicable; ii) As we enhance the model's capabilities and increase the volume of training data, up to a certain point, the gap becomes sufficiently small, allowing these emergent abilities to commence; iii) Beyond that, the utilization of even more data and increasingly capable models enhances the manifestation of the emergent abilities in Wei et al. (2022b) as it well approaches the asymptotic results in our paper. In essence, our theoretical findings imply the scaling behavior of LLMs and suggest that the emergence of these abilities is contingent upon the scale of the model and the amount of training data employed.

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

# Appendix

## A  PROOF OF PROPOSITION 1

For each massage $\mathbf{x}_1$, $\mathbf{x}_2$, we have

$$\Pr(\theta_1 = \theta_*|\mathbf{x}_1) \geq 1 - \varepsilon_1 \implies \sum_{\theta_1 \neq \theta_*} \frac{\Pr(\mathbf{x}_1, \theta_1)}{\Pr(\mathbf{x}_1)} \leq \varepsilon_1$$

$$\Pr(\theta_2 = \theta_*|\mathbf{x}_2) \geq 1 - \varepsilon_2 \implies \sum_{\theta_2 \neq \theta_*} \frac{\Pr(\mathbf{x}_2, \theta_2)}{\Pr(\mathbf{x}_2)} \leq \varepsilon_2$$

When we concatenate two independent messages $\mathbf{x}_1$ and $\mathbf{x}_2$ into a single composite message $(\mathbf{x}_1, \mathbf{x}_2)$. We treat it as one message rather two separate ones, and meanwhile we only deduce a single intention out of it (rather than two separate intentions). Therefore, this allows us to consider a sub-space in the latent space where the intentions of these two messages are tied together.

When we deduce the common intention $\theta_*$ from the composite message $(\mathbf{x}_1, \mathbf{x}_2)$, we have

$$
\begin{aligned}
\Pr(\theta_1 = \theta_*, \theta_2 = \theta_*|\mathbf{x}_1, \mathbf{x}_2) &= 1 - \sum_{\lambda \neq \theta_*} \Pr(\theta_1 = \lambda, \theta_2 = \lambda|\mathbf{x}_1, \mathbf{x}_2) \\
&= 1 - \sum_{\lambda \neq \theta_*} \frac{\Pr(\mathbf{x}_1, \theta_1 = \lambda, \mathbf{x}_2, \theta_2 = \lambda)}{\Pr(\mathbf{x}_1, \mathbf{x}_2)} \\
&= 1 - \sum_{\lambda \neq \theta_*} \frac{\Pr(\mathbf{x}_1, \theta_1 = \lambda)\Pr(\mathbf{x}_2, \theta_2 = \lambda)}{\Pr(\mathbf{x}_1)\Pr(\mathbf{x}_2)} \\
&\geq 1 - \Big(\sum_{\theta_1 \neq \theta_*} \frac{\Pr(\mathbf{x}_1, \theta_1)}{\Pr(\mathbf{x}_1)}\Big)\Big(\sum_{\theta_2 \neq \theta_*} \frac{\Pr(\mathbf{x}_2, \theta_2)}{\Pr(\mathbf{x}_2)}\Big) \\
&\geq 1 - \varepsilon_1 \varepsilon_2
\end{aligned}
$$

## B  THE SPARSITY PROPERTY OF THE JOINT DISTRIBUTION OF LANGUAGES

For any unambiguous language, we have

$$
\begin{aligned}
\Pr(\theta_0|\mathbf{x}) = 1 \quad &\implies \quad \Pr(\theta_0|\mathbf{x}) = \frac{q(\theta_0, \mathbf{x})}{q(\mathbf{x})} = 1 \\
&\implies \quad q(\mathbf{x}) = q(\theta_0, \mathbf{x}) \\
&\implies \quad q(\theta, \mathbf{x}) = \begin{cases} q(\mathbf{x}) & \theta = \theta_0 \\ 0 & \theta \neq \theta_0 \end{cases}
\end{aligned}
$$

For an $\varepsilon$-ambiguous language, we have

$$
\begin{aligned}
\Pr(\theta_0|\mathbf{x}) \geq 1 - \varepsilon(\mathbf{x}) \quad &\implies \quad \Pr(\theta_0|\mathbf{x}) = \frac{q(\theta_0, \mathbf{x})}{q(\mathbf{x})} \geq 1 - \varepsilon(\mathbf{x}) \\
&\implies \quad q(\theta_0, \mathbf{x}) \geq \big(1 - \varepsilon(\mathbf{x})\big)q(\mathbf{x}) \\
&\implies \quad q(\theta_0, \mathbf{x}) \geq \big(1 - \varepsilon(\mathbf{x})\big)\big(q(\theta_0, \mathbf{x}) + q(\Theta\backslash\theta_0, \mathbf{x})\big) \\
&\implies \quad \frac{q(\theta_0, \mathbf{x})}{q(\Theta\backslash\theta_0, \mathbf{x})} \geq \frac{1 - \varepsilon(\mathbf{x})}{\varepsilon(\mathbf{x})} \approx \frac{1}{\varepsilon(\mathbf{x})}
\end{aligned}
$$

where $\Theta\backslash\theta_0$ denotes a complement set containing all elements in $\Theta$ except $\theta_0$, and $q(\Theta\backslash\theta_0, \mathbf{x})$ is defined as the total residue after the contribution of the intention $\theta_0$ is taken out:

$$q(\Theta\backslash\theta_0, \mathbf{x}) \triangleq \sum_{\theta \in \Theta, \theta \neq \theta_0} q(\theta, \mathbf{x}) = q(\mathbf{x}) - q(\theta_0, \mathbf{x})$$

## C  PROOF OF THEOREM 1

As the consistency is a well-accepted property of maximum likelihood estimation (MLE), the result in Theorem 1 should not be too surprised.

Firstly, due to that the model $p_\Lambda(\mathbf{x})$ is a universal density approximator, for any given distribution $q(\mathbf{x})$, there exists a set of model parameters, denoted as $\Lambda_*$, to make $p_{\Lambda_*}(\mathbf{x}) = q(\mathbf{x})$ hold for all $\mathbf{x}$.

Secondly, we define an expected log-likelihood function as follows:

$$l(\Lambda) = \int_\mathbf{x} q(\mathbf{x}) \ln p_\Lambda(\mathbf{x})d\mathbf{x} = \int_\mathbf{x} p_{\Lambda_*}(\mathbf{x}) \ln p_\Lambda(\mathbf{x})d\mathbf{x}$$

Based on Jensen's inequality, for any $\Lambda \neq \Lambda_*$, we have $l(\Lambda) \leq l(\Lambda_*)$ because of

$$\int_\mathbf{x} p_{\Lambda_*}(\mathbf{x}) \ln p_\Lambda(\mathbf{x})d\mathbf{x} - \int_\mathbf{x} p_{\Lambda_*}(\mathbf{x}) \ln p_{\Lambda_*}(\mathbf{x})d\mathbf{x} \leq 0$$

Therefore, $\Lambda_*$ is the maximum point of the above function $l(\Lambda)$, i.e. $l(\Lambda_*) = \max_\Lambda l(\Lambda)$.

Thirdly, the maximum likelihood estimation $\Lambda_n$ is the maximum point of the following empirical log-likelihood function:

$$l_n(\Lambda) = \frac{1}{n} \sum_{i=1}^{n} \ln p_\Lambda(\mathbf{x}_i)$$

where $\{\mathbf{x}_1, \mathbf{x}_2, \cdots, \mathbf{x}_n\}$ are $n$ i.i.d. samples randomly drawn from the distribution $p_{\Lambda_*}(\mathbf{x})$. According to the law of large numbers, we have

$$\lim_{n \to \infty} l_n(\Lambda) = l(\Lambda)$$

As a result, the maximum value of $l_n(\Lambda)$ converges to the maximum value of $l(\Lambda)$. That is

$$\lim_{n \to \infty} \max_\Lambda l_n(\Lambda) = \max_\Lambda l(\Lambda) = l(\Lambda_*)$$

At last, because neural networks, including transformer models, usually do not satisfy the identification requirement, we can not immediately conclude $\Lambda_n \to \Lambda_*$ as $n \to \infty$. Following the technique in Jiang (2019), under the condition of over-parameterization, the model can be mapped to a canonical model space where the identification requirement holds. By doing that, we conclude that if the model is sufficiently large, some simple optimization methods, such as stochastic gradient descent, will ensure that $\Lambda_n$ converges to a point that achieves the same objective function values as $l(\Lambda_*)$. In other words, it converges to a model that is at least as good as $\Lambda_*$.

## D  PROOF OF PROPOSITION 2

We first start with the conditional distribution of the LLM $\boldsymbol{\Lambda}_*$ as:

$$\begin{aligned} p_{\boldsymbol{\Lambda}_*}(\mathbf{y}|\mathbf{x}) &= \frac{p_{\boldsymbol{\Lambda}_*}(\mathbf{x}, \mathbf{y})}{p_{\boldsymbol{\Lambda}_*}(\mathbf{x})} = \frac{q(\mathbf{x}, \mathbf{y})}{q(\mathbf{x})} \\ &= \frac{q(\mathbf{x}, \theta_\mathbf{x}, \mathbf{y}) + \sum_{\theta \neq \theta_\mathbf{x}} q(\mathbf{x}, \theta, \mathbf{y})}{q(\mathbf{x}, \theta_\mathbf{x}) + \sum_{\theta \neq \theta_\mathbf{x}} q(\mathbf{x}, \theta)} \\ &= \frac{q(\mathbf{x}, \theta_\mathbf{x}, \mathbf{y}) + \sum_{\theta \neq \theta_\mathbf{x}} q(\mathbf{x}, \theta)q(\mathbf{y}|\mathbf{x}, \theta)}{q(\mathbf{x}, \theta_\mathbf{x}) + \sum_{\theta \neq \theta_\mathbf{x}} q(\mathbf{x}, \theta)} \end{aligned}$$

Due to $0 \leq q(\mathbf{y}|\mathbf{x}, \theta)) \leq 1$, we have

$$\frac{q(\mathbf{x}, \theta_\mathbf{x}, \mathbf{y})}{q(\mathbf{x}, \theta_\mathbf{x}) + \sum_{\theta \neq \theta_\mathbf{x}} q(\mathbf{x}, \theta)} \leq p_{\boldsymbol{\Lambda}_*}(\mathbf{y}|\mathbf{x}) \leq \frac{q(\mathbf{x}, \theta_\mathbf{x}, \mathbf{y}) + \sum_{\theta \neq \theta_\mathbf{x}} q(\mathbf{x}, \theta)}{q(\mathbf{x}, \theta_\mathbf{x}) + \sum_{\theta \neq \theta_\mathbf{x}} q(\mathbf{x}, \theta)}$$

If we denote $\delta = \frac{\sum_{\theta \neq \theta_\mathbf{x}} q(\mathbf{x}, \theta)}{q(\mathbf{x}, \theta_\mathbf{x})}$, then we have

$$\frac{q(\mathbf{y}|\mathbf{x}, \theta_\mathbf{x})}{1 + \delta} \leq p_{\boldsymbol{\Lambda}_*}(\mathbf{y}|\mathbf{x}) \leq \frac{q(\mathbf{y}|\mathbf{x}, \theta_\mathbf{x}) + \delta}{1 + \delta}$$

According to the sparsity property of $\varepsilon$-ambiguous languages, we have $0 \leq \delta \leq \varepsilon(\mathbf{x})$. As a result, we have

$$\frac{q(\mathbf{y}|\mathbf{x}, \theta_{\mathbf{x}})}{1 + \varepsilon(\mathbf{x})} \leq p_{\mathbf{\Lambda}_*}(\mathbf{y}|\mathbf{x}) \leq q(\mathbf{y}|\mathbf{x}, \theta_{\mathbf{x}}) + \varepsilon(\mathbf{x})$$

Due to $\frac{1}{1+\varepsilon} > 1 - \varepsilon$, we have

$$\left| p_{\mathbf{\Lambda}_*}(\mathbf{y}|\mathbf{x}) - q(\mathbf{y}|\mathbf{x}, \theta_{\mathbf{x}}) \right| \leq \varepsilon(\mathbf{x})$$

## E    PROOF OF PROPOSITION 3

Here we assume the language generation is unambiguous. Let's consider how the LLMs will compute the following conditional probability given the prompt $\mathbf{x}$, all examples $\{\mathbf{i}_k, \mathbf{o}_k \,|\, k = 1, 2, \cdots, m\}$, and an new input $\mathbf{i}_{m+1}$ for the same task. Note that all these conditions are generated under the same intention but they are independent as they are not generated by conditioning each other. However, the reply $\mathbf{y}$ is not independent from these conditions because $\mathbf{y}$ is generated conditioning on all of them.

$$
\begin{aligned}
&p_{\mathbf{\Lambda}_*}(\mathbf{y}|\mathbf{x}, \mathbf{i}_1, \mathbf{o}_1, \cdots, \mathbf{i}_m, \mathbf{o}_m, \mathbf{i}_{m+1}) \\
&= \frac{p_{\mathbf{\Lambda}_*}(\mathbf{x}, \mathbf{i}_1, \mathbf{o}_1, \cdots, \mathbf{i}_m, \mathbf{o}_m, \mathbf{i}_{m+1}, \mathbf{y})}{p_{\mathbf{\Lambda}_*}(\mathbf{x}, \mathbf{i}_1, \mathbf{o}_1, \cdots, \mathbf{i}_m, \mathbf{o}_m, \mathbf{i}_{m+1})} \\
&= \frac{q(\mathbf{x}, \mathbf{i}_1, \mathbf{o}_1, \cdots, \mathbf{i}_m, \mathbf{o}_m, \mathbf{i}_{m+1}, \mathbf{y})}{q(\mathbf{x}, \mathbf{i}_1, \mathbf{o}_1, \cdots, \mathbf{i}_m, \mathbf{o}_m, \mathbf{i}_{m+1})} \\
&= \frac{\sum_{\theta_0, \theta_1, \cdots, \theta_m} q(\mathbf{x}, \theta_0, \mathbf{i}_1, \mathbf{o}_1, \theta_1, \cdots, \mathbf{i}_m, \mathbf{o}_m, \theta_m, \mathbf{i}_{m+1}, \mathbf{y})}{\sum_{\theta_0, \theta_1, \cdots, \theta_m} q(\mathbf{x}, \theta_0, \mathbf{i}_1, \mathbf{o}_1, \theta_1, \cdots, \mathbf{i}_m, \mathbf{o}_m, \theta_m, \mathbf{i}_{m+1})} \\
&= \frac{\sum_{\theta_0, \theta_1, \cdots, \theta_m} q(\mathbf{x}, \theta_0) \prod_{k=1}^m q(\mathbf{i}_k, \mathbf{o}_k, \theta_k) q(\mathbf{i}_{m+1}, \mathbf{y}|\theta_0, \cdots, \theta_m)}{\sum_{\theta_0, \theta_1, \cdots, \theta_m} q(\mathbf{x}, \theta_0) \prod_{k=1}^m q(\mathbf{i}_k, \mathbf{o}_k, \theta_k) q(\mathbf{i}_{m+1}|\theta_0, \cdots, \theta_m)} \\
&= \frac{q(\mathbf{x}, \theta_*) \prod_{k=1}^m q(\mathbf{i}_k, \mathbf{o}_k, \theta_*) q(\mathbf{i}_{m+1}, \mathbf{y}|\theta_*)}{q(\mathbf{x}, \theta_*) \prod_{k=1}^m q(\mathbf{i}_k, \mathbf{o}_k, \theta_*) q(\mathbf{i}_{m+1}|\theta_*)} = \frac{p(\mathbf{i}_{m+1}, \mathbf{y}|\theta_*)}{q(\mathbf{i}_{m+1}|\theta_*)} \\
&= q(\mathbf{y}|\mathbf{i}_{m+1}, \theta_*)
\end{aligned}
$$

At the above, we have applied the sparsity property of unambiguous languages in eq.(3) because the prompt and all examples are generated under the same intention $\theta_*$. For all other intentions $\theta \neq \theta_*$, we have $q(\mathbf{x}, \theta) = 0$, and $q(\mathbf{i}_k, \mathbf{o}_k, \theta) = 0$, and $q(\mathbf{i}_{k+1}, \theta) = 0$.

## F    PROOF OF PROPOSITION 4

Under the ICL setting, we assume all provided examples are created independently for a common intention $\theta_*$, which is the same as the initial instruction text $\mathbf{x}$. As a result, we only need to consider a subspace of the latent space, where all underlying intentions are tied together.

$$
\begin{aligned}
&q(\mathbf{x}, \mathbf{i}_1, \mathbf{o}_1, \cdots, \mathbf{i}_m, \mathbf{o}_m, \mathbf{i}_{m+1}) \\
&= \sum_{\lambda \in \Theta} q(\mathbf{x}, \theta_0 = \lambda, \mathbf{i}_1, \mathbf{o}_1, \theta_1 = \lambda, \cdots, \mathbf{i}_m, \mathbf{o}_m, \theta_m = \lambda, \mathbf{i}_{m+1}, \theta_{m+1} = \lambda)
\end{aligned}
$$

For $\varepsilon$-ambiguous languages, as in Appendix E, we can derive:

$$p_{\Lambda_*}(\mathbf{y}|\mathbf{x}, \mathbf{i}_1, \mathbf{o}_1, \cdots, \mathbf{i}_m, \mathbf{o}_m, \mathbf{i}_{m+1})$$

$$= \frac{\sum_{\lambda \in \Theta} q\big(\mathbf{x}, \theta_0 = \lambda, \mathbf{i}_1, \mathbf{o}_1, \theta_1 = \lambda, \cdots, \mathbf{i}_m, \mathbf{o}_m, \theta_m = \lambda, \mathbf{i}_{m+1}, \theta_{m+1} = \lambda, \mathbf{y}\big)}{\sum_{\lambda \in \Theta} q\big(\mathbf{x}, \theta_0 = \lambda, \mathbf{i}_1, \mathbf{o}_1, \theta_1 = \lambda, \cdots, \mathbf{i}_m, \mathbf{o}_m, \theta_m = \lambda, \mathbf{i}_{m+1}, \theta_{m+1} = \lambda\big)}$$

$$= \frac{q\big(\mathbf{x}, \theta_*, \mathbf{i}_1, \mathbf{o}_1, \theta_*, \cdots, \mathbf{i}_{m+1}, \theta_*, \mathbf{y}\big) + \sum_{\lambda \neq \theta_*} q\big(\mathbf{x}, \theta_0 = \lambda, \mathbf{i}_1, \mathbf{o}_1, \theta_1 = \lambda, \cdots, \mathbf{i}_m, \mathbf{o}_m, \theta_m = \lambda, \mathbf{i}_{m+1}, \theta_{m+1} = \lambda, \mathbf{y}\big)}{q\big(\mathbf{x}, \theta_*, \mathbf{i}_1, \mathbf{o}_1, \theta_*, \cdots, \mathbf{i}_{m+1}, \theta_*\big) + \sum_{\lambda \neq \theta_*} q\big(\mathbf{x}, \theta_0 = \lambda, \mathbf{i}_1, \mathbf{o}_1, \theta_1 = \lambda, \cdots, \mathbf{i}_m, \mathbf{o}_m, \theta_m = \lambda, \mathbf{i}_{m+1}, \theta_{m+1} = \lambda\big)}$$

$$\tag{12}$$

$$\leq \frac{q\big(\mathbf{x}, \theta_*, \mathbf{i}_1, \mathbf{o}_1, \theta_*, \cdots, \mathbf{i}_{m+1}, \theta_*, \mathbf{y}\big) + \sum_{\lambda \neq \theta_*} q\big(\mathbf{x}, \theta_0 = \lambda, \mathbf{i}_1, \mathbf{o}_1, \theta_1 = \lambda, \cdots, \mathbf{i}_m, \mathbf{o}_m, \theta_m = \lambda, \mathbf{i}_{m+1}, \theta_{m+1} = \lambda\big)}{q\big(\mathbf{x}, \theta_*, \mathbf{i}_1, \mathbf{o}_1, \theta_*, \cdots, \mathbf{i}_{m+1}, \theta_*\big) + \sum_{\lambda \neq \theta_*} q\big(\mathbf{x}, \theta_0 = \lambda, \mathbf{i}_1, \mathbf{o}_1, \theta_1 = \lambda, \cdots, \mathbf{i}_m, \mathbf{o}_m, \theta_m = \lambda, \mathbf{i}_{m+1}, \theta_{m+1} = \lambda\big)}$$

$$= \frac{q(\mathbf{y}|\mathbf{i}_{m+1}, \theta_*) + \overbrace{\dfrac{\sum_{\lambda \neq \theta_*} q\big(\mathbf{x}, \theta_0 = \lambda, \mathbf{i}_1, \mathbf{o}_1, \theta_1 = \lambda, \cdots, \mathbf{i}_m, \mathbf{o}_m, \theta_m = \lambda, \mathbf{i}_{m+1}, \theta_{m+1} = \lambda\big)}{q\big(\mathbf{x}, \theta_*, \mathbf{i}_1, \mathbf{o}_1, \theta_*, \cdots, \mathbf{i}_m, \mathbf{o}_m, \theta_*, \mathbf{i}_{m+1}\big)}}^{=\eta}}{1 + \underbrace{\dfrac{\sum_{\lambda \neq \theta_*} q\big(\mathbf{x}, \theta_0 = \lambda, \mathbf{i}_1, \mathbf{o}_1, \theta_1 = \lambda, \cdots, \mathbf{i}_m, \mathbf{o}_m, \theta_m = \lambda, \mathbf{i}_{m+1}, \theta_{m+1} = \lambda\big)}{q\big(\mathbf{x}, \theta_*, \mathbf{i}_1, \mathbf{o}_1, \theta_*, \cdots, \mathbf{i}_m, \mathbf{o}_m, \theta_*, \mathbf{i}_{m+1}\big)}}_{=\eta}}$$

At above, we have applied the following inequality:

$$\sum_{\lambda \neq \theta_*} q\big(\mathbf{x}, \theta_0 = \lambda, \mathbf{i}_1, \mathbf{o}_1, \theta_1 = \lambda, \cdots, \mathbf{i}_m, \mathbf{o}_m, \theta_m = \lambda, \mathbf{i}_{m+1}, \theta_{m+1} = \lambda, \mathbf{y}\big)$$

$$= \sum_{\lambda \neq \theta_*} q\big(\mathbf{x}, \theta_0 = \lambda, \mathbf{i}_1, \mathbf{o}_1, \theta_1 = \lambda, \cdots, \mathbf{i}_m, \mathbf{o}_m, \theta_m = \lambda, \mathbf{i}_{m+1}, \theta_{m+1} = \lambda\big) q(\mathbf{y}|\theta_0, \cdots, \theta_{m+1})$$

$$\leq \sum_{\lambda \neq \theta_*} q\big(\mathbf{x}, \theta_0 = \lambda, \mathbf{i}_1, \mathbf{o}_1, \theta_1 = \lambda, \cdots, \mathbf{i}_m, \mathbf{o}_m, \theta_m = \lambda, \mathbf{i}_{m+1}, \theta_{m+1} = \lambda\big) \qquad [\text{ due to } q(\mathbf{y}|\theta_0, \cdots, \theta_{m+1}) \leq 1]$$

We further consider $\eta$:

$$0 \leq \eta \;=\; \frac{\sum_{\lambda \neq \theta_*} q\big(\mathbf{x}, \theta_0 = \lambda, \mathbf{i}_1, \mathbf{o}_1, \theta_1 = \lambda, \cdots, \mathbf{i}_m, \mathbf{o}_m, \theta_m = \lambda, \mathbf{i}_{m+1}, \theta_{m+1} = \lambda\big)}{q\big(\mathbf{x}, \theta_*, \mathbf{i}_1, \mathbf{o}_1, \theta_*, \cdots, \mathbf{i}_m, \mathbf{o}_m, \theta_*, \mathbf{i}_{m+1}\big)}$$

$$=\; \frac{\sum_{\lambda \neq \theta_*} q(\mathbf{x}, \theta_0) \prod_{k=1}^m q(\mathbf{i}_k, \mathbf{o}_k, \theta_k) q(\mathbf{i}_{m+1}, \theta_{m+1})}{q(\mathbf{x}, \theta_*) \prod_k^m q(\mathbf{i}_k, \mathbf{o}_k, \theta_*) q(\mathbf{i}_{m+1}, \theta_*)}$$

$$\leq\; \frac{\sum_{\theta_0 \neq \theta_*} q(\mathbf{x}, \theta_0)}{q(\mathbf{x}, \theta_*)} \prod_{k=1}^m \frac{\sum_{\theta_k \neq \theta_*} q(\mathbf{i}_k, \mathbf{o}_k, \theta_k)}{q(\mathbf{i}_k, \mathbf{o}_k, \theta_*)} \frac{\sum_{\theta_{m+1} \neq \theta_*} q(\mathbf{i}_{m+1}, \theta_{m+1})}{q(\mathbf{i}_{m+1}, \theta_*)}$$

$$\leq\; \varepsilon(\mathbf{x}) \varepsilon(\mathbf{i}_{m+1}) \prod_{k=1}^m \varepsilon(\mathbf{i}_k, \mathbf{o}_k) \leq \varepsilon_0^{m+2}$$

where $\varepsilon_0$ denotes the maximum ambiguity among the prompt and all examples.

On the other hand, due to

$$\sum_{\lambda \neq \theta_*} q\big(\mathbf{x}, \theta_0 = \lambda, \mathbf{i}_1, \mathbf{o}_1, \theta_1 = \lambda, \cdots, \mathbf{i}_{m+1}, \theta_{m+1} = \lambda, \mathbf{y}\big) \geq 0$$

we can derive the lower bound of eq.(12) as follows:

$$p_{\Lambda_*}(\mathbf{y}|\mathbf{x}, \mathbf{i}_1, \mathbf{o}_1, \cdots, \mathbf{i}_m, \mathbf{o}_m, \mathbf{i}_{m+1}) \geq \frac{q(\mathbf{y}|\mathbf{i}_{m+1}, \theta_*)}{1 + \eta}$$

Similar to the last step in Appendix D, for $\varepsilon$-ambiguous languages, we finally have

$$\left| p_{\Lambda_*}(\mathbf{y}|\mathbf{x}, \mathbf{i}_1, \mathbf{o}_1, \cdots, \mathbf{i}_m, \mathbf{o}_m, \mathbf{i}_{m+1}) - q(\mathbf{y}|\mathbf{i}_{m+1}, \theta_*) \right| \leq \varepsilon(\mathbf{x}) \varepsilon(\mathbf{i}_{m+1}) \prod_{k=1}^m \varepsilon(\mathbf{i}_k, \mathbf{o}_k) \leq \varepsilon_0^{m+2}$$

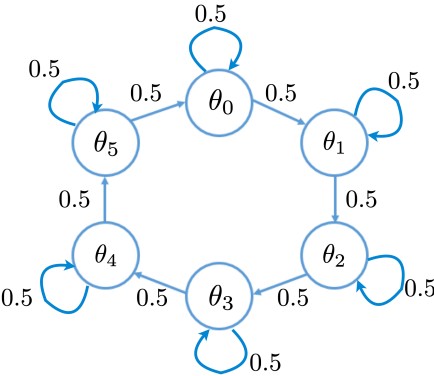

Figure 3: A 6-state circular Markov chain model is used to simulate the prior intention distribution $q(\theta)$ in the latent space model for language generation.

## G   MORE DETAILS ON EXPERIMENTAL SETTINGS

### G.1   DATA GENERATION

We use a 6-state circular Markov chain, as shown in Figure 3, to model the prior distribution of intentions, i.e. $q(\theta)$, in the step one of the latent space model for language generation. Each state corresponds to a distinct intention. As a result, the latent space $\Theta$ consists of only 6 elements.

Given each intention, we use another intention-specific Markov chain to generate a message for it. These intention-dependent Markov chains consists of 18 states, corresponding to 18 alphabetic letters (from $a$ to $r$). To generate unambiguous languages, we use only three distinct letters to construct a message for each intention, i.e. only using $\{a, b, c\}$ for $\theta_0$, $\{d, e, f\}$ for $\theta_1$, $\{g, h, i\}$ for $\theta_2$, $\{j, k, l\}$ for $\theta_3$, $\{m, n, o\}$ for $\theta_4$, and $\{p, q, r\}$ for $\theta_5$. In this case, each intention-dependent Markov chain has only $3 \times 3$ nonzero probabilities (corresponding to the three selected letters) in its $18 \times 18$ transition matrix. On the other hand, to generate $\varepsilon$-ambiguous languages, we inject small amounts of noise into these sparse transition matrices so that each letter has a small probability of jumping to other letters rather than those dedicated to the same intention. By varying the amount of noise added to the transition matrices, we can simulate different levels of ambiguity. After generating a message for a given intention using an intention-dependent Markov chain, a newline character is appended to signify the message's end. For the sake of simplicity, we generate all messages under the assumption that every message has a consistent length, comprised of 20 letters and one newline character. After each message is generated, using the circular Markov chain, the next intention will be selected, and another message will be generated for that intention in the same way. This process can be repeated as many times as desired to generate any number of messages.

Some unambiguous messages generated in this way are shown below as an example:

```
...
qqpqrqrqqqqrrqprqqqqq
abccbacbaabbbbcaaacba
effeffeefffffffffeffff
feffffffeddddfeeffffff
deeedfedeffeedffffffff
hgiiiiiiigihgihgiihih
jjjklkjjkjkjkjkllkkkk
kjkkkllkjkjjjklllljkkk
omnoononmnononommonom
nonoonomnnoonomnononm
rrqprrrqpqrqqprqrrrqp
baaabaaacacaabaaaaaac
baacaacbaaaaabaabbaca
caaacbabbacaaccaaabca
```

```
cabacbcabbaaccccacaaa
...
```

Some generated ambiguous messages ($\varepsilon = 0.06$) are shown below as an example:

```
...
qqqqqqafdrqqqppribmmf
aagaaaraaojogikfdjone
aaahdnbabcadnkdggkkep
cmjjfdiqhaacmbcnpaajr
deeffedelokhhhdffemef
iiiiihhghcoqqcpbliiig
ghigkabkecemgjkeefbii
iihqbrghghhhghghkmnpn
kfgoaqdgholkljkhondjc
oblohjfnnmnanhjbhonom
nfkpjgqoomaibpragedak
qrqrrprqhehdloajilopq
rlljmprrqqrrhgrrqqqpr
rqhijkpjmkcmlfeicopnr
...
```

Finally, some more ambiguous messages ($\varepsilon = 0.11$) are shown below as an example:

```
...
diporcqddofiffeffdqnc
fafnakjcdmcdbfeoqaeff
dedfepnddedenpaghceer
ededlapbnpipnirkpqqan
eialkcpfefopqojpmrffe
ffeddefcaaffipkbcohak
dffederonkqldeemkaclp
djhfpbrbjpkpcgnjhamke
efjnhlhdeddeffeeeffoi
ebchiqcjfbrihlqipdedd
gholgghgilchgjrobmrrk
lkqdhfdriqjklkkjlfbmm
lcjkjkllljllllkkjlkihn
lklnkepaojjjpqaarbrpa
lnrhmpbpmchhhdampekjj
llklkkjkrcpbliqhaaqgh
jbbklkjjllcbijkkkerge
...
```

## G.2   MODEL TRAINING

In our experiments, we train a character-level GPT-like model (Radford et al., 2018), similar to the nanoGPT model[2], on the generated synthetic languages using the ADAM algorithm on a GTX1080 Ti GPU with 12GB memory. During the training process, the learning rates are automatically decayed according to a preset annealing schedule. All training hyper-parameters for the GPT model structure and the optimizer settings are given in Table 1.

## H   LIMITATIONS

The paper's primary theoretical findings hinge on two fundamental assumptions about languages: i) they are not developed haphazardly but rather serve the purpose of conveying information; and ii) all languages are generated from a unified latent intention space, even though the explicit construction

---

[2]nanoGPT: https://github.com/karpathy/nanoGPT

| model structure | |
|---|---|
| number of layers | 3 |
| number of head | 12 |
| head size | 64 |
| block size | 32, 64, or 128 |
| number of model parameters | 21.25 million |
| optimizer | |
| initial learning rate | $6.0 \times 10^{-4}$ |
| minibatch size | 12 |
| $\beta_1$ | 0.9 |
| $\beta_2$ | 0.95 |
| maximum iterations | 5000 |

Table 1: All hyperparameters used in the training process for a GPT-like language model.

of this intention space is unknown. Compared to prior research on similar topics, such as Xie et al. (2022); Wang et al. (2023); Wies et al. (2023), our theoretical results require fewer assumptions while also offering a more comprehensive explanation of the emergent capabilities of large language models (LLMs), beyond just in-context learning.

The simulation experiments in this paper are somewhat constrained in that they only utilize a single type of $\varepsilon$-language, which is synthesized using a doubly-embedded Markov chain model. While the simulation results for this synthetic language do confirm the theoretical findings outlined in the paper, it would be worthwhile to explore other types of $\varepsilon$-languages in order to further evaluate the proposed theory.

