# OpenReview forum: "A Latent Space Theory for Emergent Abilities in Large Language Models"
_ICLR.cc/2024/Conference — Submitted to ICLR 2024_

### Official Review · Reviewer_Zr2w · 2023-10-31

**Soundness:** 3 good
**Presentation:** 3 good
**Contribution:** 3 good
**Rating:** 6
**Confidence:** 3

**Summary:**

The authors propose a model to describe capabilities from large language models (LLM). The model is based on Bayesian approaches to define the joint distribution of languages and the relation to LLM.  The main contributions are: i) modelling of LLM capabilities , and ii) classification of types of languages processed given a LLM.

**Strengths:**

- The proposed model describes capabilities of LLM in a principle manner.
- Clear description of background knowledge and related work needed to understand the proposed approach.
- The authors perform a  comparison of the proposed model on different LLM applications/capabilities, such as, in-context-learning, and chain-of-though.

**Weaknesses:**

- It is not clearly described the implication of using synthetic data on the findings.
- An extra contribution could be to present a statistical significance test or uncertainty estimates of the results.

**Questions:**

Please address the following questions during the rebuttal:

- Please elaborate on the hyperparameter selection for the LMs  and the type of LM and the impact on the proposed model.
- Is there a difference between character based model and a bpe LM on the findings?
- Please speculate on the use of natural language with the proposed study, how well the model findings will adapt to natural language?

**Details Of Ethics Concerns:**

I have no concerns.

---

### Official Review · Reviewer_5EvL · 2023-10-31

**Soundness:** 3 good
**Presentation:** 2 fair
**Contribution:** 2 fair
**Rating:** 3
**Confidence:** 4

**Summary:**

This paper aims to provide a theoretical explanation for key emergent abilities of LLMs (in-context learning, chain-of-thought prompting, instruction fine-tuning), in terms of Bayesian inference on latent intentions underlying the language.

**Strengths:**

1. This paper provides a new look at the role of the link between language and underlying intentions in enabling emergent properties. It does so using both theoretical arguments and experiments.

2. The formal results are sound, as far as I could tell (though with some question, as described below).

**Weaknesses:**

1. It remains unclear how the proposed account substantially improves over Xie et al 2022, which already explained ICL in terms of the recovery of an underlying state / intention. The overall claim, and the explanation of in-context learning in Section 5, are similar to Xie et al. 2022.
The paper additionally provides sections about Chain-of-Thought prompting (Section 6) and instruction finetuning (Section 7), but they are quite informal and unspecific.
The arguments in Section 6 are informal, using the idea that direct transitions between query and answer intentions may be rare, and that intermediate steps are more well-represented. It is left open how these ideas would be formalized and what precise assumptions are needed. In particular, as Sections 2&3 only assumed single intentions for the generative model underlying language (one intention is sampled and mapped to a text), it is unclear how the ideas in Section 6 (which refer to chains of intentions) can be formalized within the overall framework.
In Section 7, the idea seems to be that instructions provide information about the intention \theta, but this argument is again not formalized. Thus, it is left somewhat unclear how the formal results present a substantial advance over Xie et al 2022.

2. Section 5 considers prompts consisting of demonstrations of input-output pairs, and assumes that these correspond to some underlying intention \theta. The results in Section 5 show that, to the extent that underlying intentions are unambiguously recoverable, a predictor can correctly continue a prompt. However, this leaves open how an LLM would acquire the ability to infer the intention from such prompts, which typically have an unnatural repetitive structure, not well represented in the training set.

**Questions:**

1. Issue with clarity: In Section 2, the definition of \epsilon-ambiguity: What is \theta_0? The condition seems to say that, whenever we sample \theta_0 and then sample x \sim q(x|\theta_0), then Pr(\theta_0|x) \geq 1-\epsilon(x) -- is this correct? Also, it seems Pr(.|.) is not defined. A plausible interpretation is that it is a Bayesian posterior for the prior q(\theta) and the likelihood  q(x|\theta_0) -- is this correct?

2. Proof of Theorem 1: The last paragraph makes reference to neural networks and SGD, whereas the theorem statement makes no reference to these (it just refers to universal approximators, without reference to architectures or training procedures). Either the theorem needs to be rephrased, or the proof should be phrased for such general universal approximators.

---

### Official Review · Reviewer_syQW · 2023-11-03

**Soundness:** 1 poor
**Presentation:** 2 fair
**Contribution:** 1 poor
**Rating:** 1
**Confidence:** 4

**Summary:**

This paper tries to "explain" the emergent abilities of LLMs from a mixture
model viewpoint. Essentially, this paper says that given the prompt or history
fed into LLM, predicting the successive tokens amounts to finding a latent
intension, a mixture model component from a statistical point of view, that
would lead to the emergent behaviors of LLMs. Some elementary theorems and
artificial experiments are provided.

**Strengths:**

The problem itself is very important.

**Weaknesses:**

I agree that the emergent behaviors of LLMs need a thorough scientific
investigation, but this paper says almost nothing about these emergent
abilities. What is important here is that LLM seem to solve a new task that is
not contained in the training data: that is called emergent. However, just
finding a latent "intention" (quite loosely defined in this paper) cannot
explain this behavior, because the new emergent intention just doesn't exist
so far. Therefore, in spite of elementary mathematical discussions in Section
4, the paragraph "In other words, ..." virtually adds nothing about the
understanding of the behaviors of LLMs.
Considering other possible explanations, such as recently developed by Arora
et al. (2023), the arguments of this paper is too trivial to be useful in
practice.

**Questions:**

Nothing.

---

### Official Review · Reviewer_aMxK · 2023-11-13

**Soundness:** 2 fair
**Presentation:** 2 fair
**Contribution:** 2 fair
**Rating:** 3
**Confidence:** 3

**Summary:**

The authors claim that emergent abilities (in-context learning, chain-of-thought prompting, instruction fine-tuning) arise due to generation being a mixture over the compositional structures in language, which are encoding as components within the underlying probability distribution of the LLM. The proofs results and contributions seem to be minor modifications compared to (Xie, 2022).

**Strengths:**

- The authors discuss important concepts that are yet to be explained (in-context learning, CoT, instruction finetuning).
- The paper is well written and straightforward to follow.

**Weaknesses:**

- While additional concepts (chain of though, instruction finetuning) are addressed in the paper compared with prior work, there does not seem to be sufficient novelty in the theoretical results. Xie, 2022 already defined a framework for explaining in-context learning as state (or intention) estimation.
- The chain of though prompts section is not rigorously defined and does not build on the intention setup for in-context learning (Sec. 2, 3, 5). The estimation of the intent for instruction fine-tuning is also not rigorously defined.
- Character based tokenizers are used in the experimentation, are these representative of the behavior for a standard LLM (i.e. BPE tokenizer)?

**Questions:**

See weaknesses.

---

> ### Author Response · Authors · 2023-11-17
> **Rebuttal Replies**
>
> **Replies to all reviewers:**
>
> - Xie et al. assume the languages are generated from an HMM process, which is quite limited and unrealistic. Our assumption on language generation (in Section 2) is dramatically different: languages can be generated from ANY stochastic process as long as they are informative, e.g. satisfying the condition (1) or (2).
>
> - Our theoretical results equally apply to both char based LLMs and BPE based ones. They both converge to the true distribution as long as a sufficient amount of training data and a large enough model are used. All the theoretical results presented in this paper are established through rigorous mathematical proofs, rendering them self-sufficient without reliance on empirical validation. The simulation results in Section 8 are included: i) to verify certain aspects of these findings; ii) to demonstrate the scaling behavior of LLMs (varying model capacity and the amount of training data) on a simple Markovian language.
>
> **To Reviewer 5EvL**:
> - $\theta$ represents a random variable in the latent intention space, while $\theta_0$ denotes the intention of $\mathbf{x}$ in eq.(1) or (2). $\Pr(.)$ stands for a probability value and $\Pr(.|.)$ for a conditional probability.
>
> - The last paragraph in Proof of Theorem 1 is not really part of the theorem itself but just some remarks to explain Theorem 1 is practically achievable.

---

### Meta-Review · Area_Chair_JjpW · 2023-12-13

**Metareview:**

This paper attempts to give a theoretical explanation for emergent abilities observed in LLMs, e.g., in-context learning, chain-of-thought prompting, instruction fine-tuning, in terms of Bayesian inference on latent intentions underlying the language. It builds on the work of Xie et. al, 2022. All reviewers raised concerns about the paper's limited novelty, problematic connection between "emergent" abilities and what is seen vs. not seen during pretraining and unclear presentation at places.

**Justification For Why Not Higher Score:**

NA

**Justification For Why Not Lower Score:**

NA

---

### Decision · Program_Chairs · 2024-01-16

Reject